# Changes in Long-Term Functional Independence in Patients with Moderate and Severe Ischemic Stroke: Comparison of the Responsiveness of the Modified Barthel Index and the Functional Independence Measure

**DOI:** 10.3390/ijerph19159612

**Published:** 2022-08-04

**Authors:** Eun Young Lee, Min Kyun Sohn, Jong Min Lee, Deog Young Kim, Yong Il Shin, Gyung Jae Oh, Yang Soo Lee, So Young Lee, Min Keun Song, Jun Hee Han, Jeong Hoon Ahn, Young Hoon Lee, Won Hyuk Chang, Soo Mi Choi, Seon Kui Lee, Min Cheol Joo, Yun Hee Kim

**Affiliations:** 1Department of Rehabilitation Medicine, Institute of Brain Science Research, Wonkwang University School of Medicine, Iksan 54538, Korea; 2Department of Rehabilitation Medicine, College of Medicine, Chungnam National University, Daejeon 35015, Korea; 3Department of Rehabilitation Medicine, Konkuk University School of Medicine, Seoul 05030, Korea; 4Department and Research Institute of Rehabilitation Medicine, Yonsei University College of Medicine, Seoul 03722, Korea; 5Department of Rehabilitation Medicine, Pusan National University Yangsan Hospital, Pusan National University School of Medicine, Yangsan 50612, Korea; 6Department of Preventive Medicine, Wonkwang University, School of Medicine, Iksan 54538, Korea; 7Department of Rehabilitation Medicine, Kyungpook National University Hospital, Kyungpook National University School of Medicine, Daegu 41566, Korea; 8Department of Rehabilitation Medicine, Jeju National University Hospital, Jeju National University School of Medicine, Jeju 63241, Korea; 9Department of Physical and Rehabilitation Medicine, Chunnam National University Medical School, Kwangju 61469, Korea; 10Department of Statistics, Hallym University, Chuncheon 24252, Korea; 11Department of Health Convergence, Ewha Womans University, Seoul 03760, Korea; 12Department of Physical and Rehabilitation Medicine, Center for Prevention and Rehabilitation, Samsung Medical Center, Sungkyunkwan University School of Medicine, Seoul 06351, Korea; 13Division of Chronic Disease Prevention, Korea Centers for Disease Control and Prevention, Center for Disease, Cheongju 28159, Korea; 14Department of Rehabilitation Medicine, Institute of Wonkwang Medical Science, Wonkwang University School of Medicine, Iksan 54538, Korea; 15Department of Health Science and Technology, SAIHST, Sungkyunkwan University, Seoul 06351, Korea

**Keywords:** functional independence measure, independence, modified Barthel Index, severity, stroke

## Abstract

This study investigated the long-term functional changes in patients with moderate-to-severe ischemic stroke. In addition, we investigated whether there was a difference between the modified Barthel Index (MBI) and Functional Independence Measure (FIM) according to severity. To evaluate the changes in the long-term functional independence of the subjects, six evaluations were conducted over 2 years, and the evaluation was performed using MBI and FIM. A total of 798 participants participated in this study, of which 673 were classified as moderate and 125 as severe. During the first 3 months, the moderate group showed greater recovery than the severe group. The period of significant change in the National Institutes of Health Stroke Scale (NIHSS) score was up to 6 months after onset in the moderate group, and up to 3 months after onset in the severe group. In the severe group, MBI evaluation showed significant changes up to 6 months after onset, whereas FIM showed significant changes up to 18–24 months. Our results showed that functional recovery of patients with ischemic stroke in the 3 months after onset was greater in the moderate group than in the severe group. FIM is more appropriate than MBI for evaluating the functional status of patients with severe stroke.

## 1. Introduction

Stroke is a major cause of mortality and morbidity in the United States. It is the second leading cause of death worldwide and a major cause of disability [1,2,3]. Ischemic stroke is the most common type of stroke and occurs when a blood vessel in the neck or brain is blocked [4]. It is estimated that 25% to 75% of the 50 million stroke survivors need help in their daily lives [5]. Such functional dependence of stroke patients can be a burden to their family members and caregivers [6]. Functional independence is a significant goal for stroke patients and a major focus in studies of stroke recovery [7,8].

Despite the neurological damage and impairment caused by stroke, most stroke patients achieve functional recovery over time [9,10]. Such functional recovery in stroke patients shows notable changes during the first 3 months after stroke onset and reaches a plateau between 3 and 6 months [11,12,13,14]. Several studies have shown that the functional status is significantly improved even after 6 months from stroke onset [15,16,17,18]. In another study, functional recovery was achieved by 12 months in 30% of stroke patients [19].

Recently, there have been major advances in the care and rehabilitation of stroke. Therefore, updated prognostic data are needed for the functional recovery of stroke patients [20]. The results of long-term functional recovery after stroke and the knowledge of such recovery periods are critical for determining proper treatment and the rehabilitation period [21,22]. Examining changes in the long-term functional recovery after 6 months from the onset of stroke may elucidate the full extent of recovery and the point of plateau where there is a much less dynamic rate.

The functional recovery of stroke patients is also associated with the initial severity of impairment [23,24]. The severity of initial neurological damage can be quantitatively measured and classified by the National Institutes of Health Stroke Scale (NIHSS) [25]. Moderate and severe stroke types exhibit a low degree and slow rate of functional recovery compared with mild stroke [26,27]. Although there have been several studies on the long-term functional recovery of stroke, studies comparing the changes in recovery according to severity are still lacking. Severe neurological damage at the initial stage of stroke has negative effects on functional recovery [28,29,30]. Therefore, changes in functional recovery should be closely monitored in patients with moderate and severe stroke.

The Barthel Index (BI) and Functional Independence Measure (FIM) are the most highly recommended tools for assessing functional recovery in patients with stroke. These tools are credible and have been proven to have high validity for measuring the functional outcome of stroke patients [31,32,33]. Glowinski and Blazejewski (2020) proposed a model for the evaluation of patient conditions and the rehabilitation progress [34]. The degree of functional changes depends on the severity of stroke, and a tool that enables a more sensitive assessment of such functional changes needs to be explored.

In this study, we aimed to examine long-term functional changes in patients with moderate and severe ischemic stroke for 2 years from onset and to determine the differences among the results according to stroke severity by comparing the modified Barthel Index (MBI) and FIM scores.

## 2. Materials and Methods

### 2.1. The Korean Stroke Cohort for Functioning and Rehabilitation (KOSCO)

KOSCO is a large multicenter prospective cohort study of patients with primary stroke who are admitted to university hospitals in nine regions of South Korea. Patients with primary stroke who were admitted to the hospital between August 2012 and June 2014 were recruited in the current study, although the KOSCO study is still ongoing. KOSCO is a 10-year longitudinal follow-up study investigating the factors that affect residual disabilities, activity limitation, and long-term quality of life in patients with first-time strokes.

The subjects were patients aged ≥19 years (the standard age for adults in Korea) who were diagnosed with primary stroke, admitted to the hospital within 7 days of onset, and met the selection criteria. Patients with a history of stroke, those diagnosed with transient ischemic attack, and those with traumatic intracerebral hemorrhage were excluded.

The patients underwent a face-to-face baseline evaluation 7 days after onset. For the assessment, the tools that could determine the functional status of stroke patients, such as the NIHSS, MBI, FIM, Korean Mini-Mental State Examination (K-MMSE), Fugl-Meyer Assessment (FMA), Modified Ashworth Scale (MAS), 9-hole pegboard test, Functional Ambulatory Category (FAC), American Speech Language Hearing Association-National Outcome Measurement System Swallowing Scale (ASHA-NOMS), Korean Frenchay Aphasia Screening Test (K-FAST), and Modified Rankin Scale (MRS) were used. Additionally, the health conditions of patients and guardians, their mood, and changes in their quality of life were assessed. After the initial baseline assessment, the subjects were evaluated at hospital discharge and 3, 6, 12, 18, and 24 months after onset. Follow-up assessments were performed each year during the study period. During this time, the assessment was continued, except for those who withdrew their consent or died. Written consent was obtained from all participants prior to the study. The study protocol was approved by the ethics committee of each hospital.

### 2.2. Study Population

During the study period, 10,636 patients were admitted to each hospital. Among them, 7858 people agreed to long-term follow-up after discharge, and 4909 people completed the follow-up evaluation up to 24 months. Among them, patients diagnosed with ischemic stroke (*n* = 3913) were selected for the study, after excluding those with hemorrhagic stroke. In addition, those who had mild stroke at admission were excluded, and only patients with moderate and severe stroke were included. Finally, 798 patients were selected for the study (Figure 1).

### 2.3. Stroke Severity and Functional Assessments

The initial stroke severity was derived from a medical record review of the NIHSS scores obtained on patient arrival at the hospital. The assessment was performed at each time point, from hospital discharge to 24 months after onset. NIHSS is a tool that can quantitatively measure the level of neurological damage in stroke patients. The total NIHSS scores were classified according to severity as mild (0–4), moderate (5–15), or severe (16–42).

MBI and FIM, which can evaluate the level of functional independence, were used in this study. The assessment was performed at each time point, from hospital discharge to 24 months after onset. The Barthel Index was first published in 1965 and consists of 10 items. Shah et al. (1989) further improved the discriminative power of the instrument by standardizing the rating criteria and scale into a five-point Likert format. The MBI, as it is known, has increased the sensitivity of the instrument both at the item and scale levels, and yielded a higher content reliability and internal consistency [33,35,36]. MBI consists of 10 daily activity items and can be analyzed using a total score ranging from 0 to 100. Each item is scored on a 5-level scale, from 1 (completely dependent) to 5 (completely independent) [33]. FIM consists of 18 daily activity items with a total score ranging from 18 to 126. Each item is rated on a 7-point scale ranging from 1 (completely dependent) to 7 (independent), according to the level of independence. A total score of ≤108 indicates the need for help from others or restrictions in activities [37].

### 2.4. Statistical Analysis

Descriptive statistics were used to analyze the demographic and clinical characteristics of the patients and the characteristics of primary stroke. Changes in MBI, FIM, and NIHSS scores in the moderate and severe groups at each assessment time point were analyzed using one-way repeated-measures ANOVA. Statistical analysis was performed using SPSS for Windows 22.0. Statistical significance was set at *p* < 0.05.

## 3. Results

### 3.1. Patients Characteristics

The final study included 798 patients according to the selection criteria. Among these 798 patients, 673 were categorized into the moderate group and 125 into the severe group. Regarding sex, there were more men (61%) in the moderate group, whereas there were more women (50.4%) in the severe group. In both groups, those who completed 12 years of education accounted for the highest percentage in the education category, and large-artery damage showed the highest percentage in the ischemia type category. The mean NIHSS scores at admission were 8.14 in the moderate group and 19.47 in the severe group. The mean K-MMSE score at admission was 20.19 in the moderate group and 14.09 in the severe group (Table 1).

### 3.2. Changes in NIHSS Scores in the Moderate and Severe Groups

Changes in the NIHSS scores were examined in the moderate and severe groups at different time points. In the moderate group, there were significant changes between discharge and 3 months after onset (*p* < 0.000), and between discharge and 3–6 months (*p* < 0.006) after onset. In the severe group, there were significant changes between discharge and 3 months after onset (*p* < 0.001) (Table 2).

### 3.3. Long-Term Changes in Functional Independence in the Moderate and Severe Groups

Changes in functional independence in patients with moderate and severe stroke over 24 months from hospital discharge were analyzed using the mean MBI and FIM scores. The results showed that the changes in the total scores of both measures were largest at 3 months after onset in both groups. In the moderate group, MBI showed improvement from 68.07 at discharge to 81.28 after onset. FIM also improved from 90.39 at discharge to 102.93 at 3 months after onset. In the severe group, MBI improved from 40.74 at discharge to 51.04 at 3 months after onset. FIM also improved from 61.33 at discharge to 69.12 at 3 months after onset (Table 3).

Functional recovery in the moderate and severe groups persisted up to 18 months after onset. The level of recovery was more significantly increased in the severe group than in the moderate group 3 months after onset. However, the severe group showed a more significant recovery from 3 to 18 months after onset. Functional recovery decreased between 18 and 24 months, with a more significant decrease in the severe group than in the moderate group (Figure 2).

### 3.4. Comparison of MBI and FIM Scores at Each Time Point in the Moderate and Severe Groups

The mean MBI and FMI scores were compared at each time point between the moderate and severe groups. The moderate group did not show any differences between the MBI and FIM scores. In both tools, significant changes were observed between discharge and 3, 3–6, and 6–12 months after onset. In the severe group, the MBI and FIM results significantly differed. MBI showed significant changes between discharge and 3 (*p* < 0.001) and 3–6 months after onset (*p* < 0.034), whereas FIM showed significant changes between discharge and 3 (*p* < 0.001), 3–6 (*p* < 0.017), 12–18 (*p* < 0.032), and 18–24 (*p* < 0.005) months after onset (Table 4).

## 4. Discussion

In this study, long-term functional changes were examined in patients with moderate and severe ischemic stroke. The moderate and severe groups were classified according to the NIHSS scores obtained at admission, and changes in functional independence were evaluated from discharge to 24 months after onset. Functional independence was assessed using MBI and FIM, and we aimed to determine the differences between the two groups.

Our results showed that functional recovery was greatest 3 months after onset in both the moderate and severe groups. In addition, the level of functional recovery at 3 months after onset was higher in the moderate group than in the severe group. One previous study also aimed to compare the neurological and functional recovery of stroke at 3 months after stroke onset after classifying stroke patients into moderate and severe groups [38]. It showed stronger recovery in the moderate group than in the severe group, which is in line with our study results. Stroke severity at onset is a valuable predictor of long-term functional recovery [39,40]. In our study, the moderate group also showed a higher level of recovery compared with the severe group because neurological recovery that occurs at the early stage after the onset of stroke is possibly affected by the size of brain lesions [41]. Neuroplasticity is the capability of stimulation by a variety of stimuli for the modulation of brain activity [42]. Brains compensate damages through reorganization and the creation of new connections among undamaged neurons [43]. After the ischemia of cells, oxygen deprivation in neurons cascades destruction in the focus of infarction being formed which lasts for many hours, usually leading to the progression of damage.

To understand the level of neurological recovery in the moderate and severe groups, the NIHSS scores at each time point were examined in this study. The results showed significant changes up to 6 months after onset in the moderate group and up to 3 months after onset in the severe group. In addition, changes in MBI scores at each time point were analyzed to understand the level of functional recovery in both groups. The moderate group showed significant changes up to 12 months after onset, whereas the severe group showed significant changes up to 6 months after onset. In both NIHSS and MBI, the moderate group showed significant changes over a longer period than the severe group. The severity of initial stroke affects the rate of recovery [44,45]. The long-term recovery observed in the moderate group compared with the severe group could be due to the relatively slower recovery rate of severe stroke; thus, it can be speculated that the level of recovery at each assessment time point was not sufficient to be statistically significant. Between 12 and 18 months after stroke onset, the level of functional recovery was higher in the severe group than in the moderate group, although the change was not statistically significant. This suggests a steady but slower and lower level of recovery in the severe group 18 months after the onset of stroke, and may be a useful guide to the most appropriate time and length of rehabilitation.

In this study, the MBI and FIM scores were compared, and the differences between the results according to the severity of stroke were examined. The results showed that the period of significant change was the same between the two assessment tools in the moderate group, whereas it was different in the severe group. A comparison of the two assessment tools showed significant changes between discharge and 3 and 3–6 months after the onset of stroke for MBI, and between discharge and 3, 3–6, 12–18, and 18–24 months after the onset of FIM. MBI is a 5-level scale and is more convenient, and clinical researchers are more familiar with it; therefore, it is more widely used in the clinical field [46,47]. However, another study reported the limited applicability of MBI to moderate and severe stroke [48]. FIM, on the other hand, does not have limited application according to the severity of disease and has more specific assessment items as a 7-level Likert scale. Therefore, it can evaluate small changes in recovery with higher sensitivity [49]. Both MBI and FIM are highly valid tools for assessing the level of functional independence in stroke patients. However, because more severe stroke shows lower and slower levels of functional recovery, our study highlights the need for an assessment tool that is more suitable for severe stroke.

Among the participants in our study, the mean score of cognitive function at admission was higher in the moderate group than in the severe group. Hence, age and cognitive function might have affected changes in functional recovery. Furthermore, the difference in the number of participants between the moderate and severe groups made it difficult to statistically compare functional recovery between the groups. In addition, the type or amount of rehabilitation patients received during hospitalization, and the location after discharge, may also affect long-term functional recovery, although this study did not consider such points. Finally, our study determined the period of long-term functional changes in patients with moderate and severe stroke; however, the activities that led to such functional recovery were not determined. Despite these limitations, our study is significant because it assessed the functional status of ischemic stroke patients through face-to-face interviews instead of phone interviews and provided data on long-term functional status during the 2 years from the onset of stroke. In addition, our study suggests a tool that allows for a more sensitive assessment of the functional status of patients with severe stroke. These data can be used for additional functional assessments of patients with ischemic stroke.

## 5. Conclusions

Our study supports previous results on the neurological recovery period of ischemic stroke patients and presents additional information on the functional recovery period. In addition, we showed that MBI and FIM, which are commonly used tools for evaluating functional recovery, enable the sensitive assessment of patients with moderate stroke, whereas FIM offers a more sensitive assessment of the functional state in patients with severe stroke. Our study results provide useful data for healthcare specialists who evaluate the functional status of stroke patients.

## Figures and Tables

**Figure 1 ijerph-19-09612-f001:**
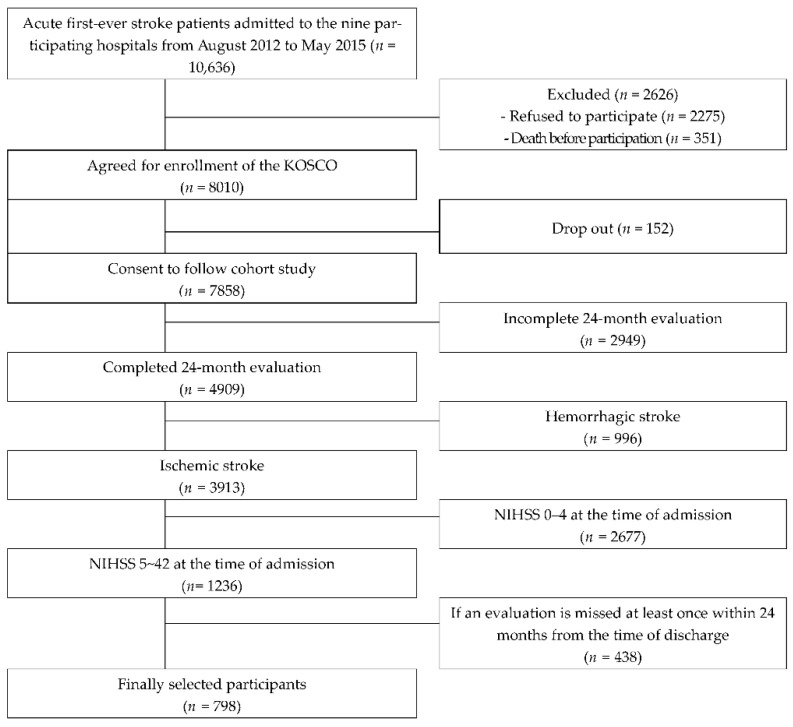
Flow diagram of the study participants.

**Figure 2 ijerph-19-09612-f002:**
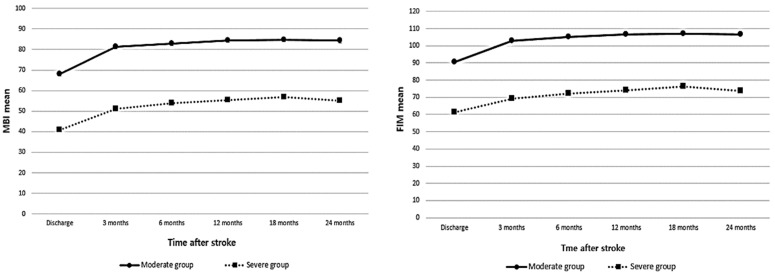
Changes in total MBI and FIM scores in moderate and severe patients.

**Table 1 ijerph-19-09612-t001:** Baseline characteristics of study population according to the severity.

		Moderate Group (*n* = 673)	Severe Group (*n* = 125)
Age (mean ± SD)		65.98 ± 12.25	69.35 ± 10.81
Sex (%)	Male	411 (61.1)	62 (49.6)
	Female	262 (38.9)	63 (50.4)
Education (%)	Uneducated	90 (13.4)	20 (16.0)
	6 years	140 (20.8)	23 (18.4)
	9 years	128 (19.0)	28 (22.4)
	12 years	196 (29.1)	30 (24.0)
	Over 14 years	90 (13.4)	20 (16.0)
	N/A	29 (4.3)	4 (3.2)
Ischemic Type (%)	Large artery atherosclerosis	376 (55.9)	65 (52.0)
	Small vessel occlusion	105 (15.6)	12 (9.6)
	Cardio-embolism	99 (14.7)	32 (25.6)
	Other determined	33 (4.9)	7 (5.6)
	Undetermined ischemic stroke	60 (8.9)	9 (7.2)
NIHSS at the time of hospitalization (mean ± SD)		8.14 ± 2.99	19.47 ± 3.60
K-MMSE at the time of hospitalization (mean ± SD)		20.19 ± 9.11	14.09 ± 11.33

Data are presented as *n* (%) or as the mean ± standard deviation. SD—standard deviation, NIHSS—National Institute of Health Stroke Scale, K-MMSE—Korean Mini-Mental State Examination.

**Table 2 ijerph-19-09612-t002:** Changes in the mean NIHSS total scores by time period for moderate and severe groups.

	Time (I)	Time (J)	MD (I-J)	SE	*p* Value	95% CI
Moderate group (*n* = 673)	Discharge	3 months	1.218 *	0.117	0.000 *	0.988–1.449
	3 months	6 months	0.229 *	0.083	0.006 *	0.065–0.397
	6 months	12 months	0.080	0.096	0.405	−0.109–0.269
	12 months	18 months	0.131	0.076	0.085	−0.018–0.280
	18 months	24 months	−0.083	0.072	0.250	−0.225–0.059
Severe group (*n* = 125)	Discharge	3 months	1.312 *	0.383	0.001 *	0.554–2.070
	3 months	6 months	0.376	0.286	0.191	−0.190–0.942
	6 months	12 months	−0.096	0.357	0.789	−0.804–0.612
	12 months	18 months	−0.008	0.311	0.979	−0.623–0.607
	18 months	24 months	−0.448	0.276	0.107	−0.994–0.098

* *p* < 0.05. MD—mean difference, SE—standard error, CI—confidence interval.

**Table 3 ijerph-19-09612-t003:** Changes in the mean of NIHSS total scores by time period for moderate and severe groups.

Time	MBI		FIM	
	Moderate Group (*n* = 673)	Severe Group (*n* = 125)	Moderate Group (*n* = 673)	Severe Group (*n* = 125)
	Mean ± SD	Mean ± SD	Mean ± SD	Mean ± SD
Discharge	68.07 ± 28.04	40.74 ± 35.62	90.39 ± 28.02	61.33 ± 36.33
3 months	81.28 ± 26.94	51.04 ± 38.09	102.93 ± 28.32	69.1 ± 38.88
6 months	82.85 ± 26.90	53.94 ± 37.76	105.04 ± 28.05	72.20 ± 38.22
12 months	84.48 ± 27.00	55.26 ± 38.94	106.57 ± 28.42	74.10 ± 39.33
18 months	84.79 ± 27.52	56.95 ± 39.45	107.02 ± 28.77	76.27 ± 40.33
24 months	84.21 ± 28.65	55.10 ± 40.22	106.80 ± 29.74	73.67 ± 41.24

MBI—modified Barthel Index, FIM—Functional Independence Measure, SD—standard deviation.

**Table 4 ijerph-19-09612-t004:** Changes in the mean MBI and FIM total scores by time period for moderate and severe groups.

		Time (I)	Time (J)	MD (J-I)	SE	*p* Value	95% CI
Moderate group (*n* = 673)	MBI	Discharge	3 months	13.214 *	0.812	0.000 *	11.6202–14.808
		3 months	6 months	1.571 *	0.405	0.000 *	0.776–2.365
		6 months	12 months	1.630 *	0.440	0.000 *	0.766–2.494
		12 months	18 months	0.308	0.367	0.402	0.413–1.028
		18 months	24 months	−0.579	0.386	0.134	−1.338–0.179
	FIM	Discharge	3 months	12.535 *	0.817	0.000 *	10.930–14.140
		3 months	6 months	2.108 *	0.435	0.000 *	1.255–2.962
		6 months	12 months	1.533 *	0.450	0.001 *	0.649–2.418
		12 months	18 months	0.447	0.381	0.241	−0.300–1.195
		18 months	24 months	−0.214	0.374	0.568	−0.949–0.521
Severe group (*n* = 125)	MBI	Discharge	3 months	10.296 *	1.467	0.000 *	7.393–13.199
		3 months	6 months	2.896 *	1.351	0.034 *	0.221–5.571
		6 months	12 months	1.328	1.106	0.232	−0.861–3.517
		12 months	18 months	1.688	1.067	0.116	−0.423–3.799
		18 months	24 months	−1.856	1.135	0.105	−4.103–0.391
	FIM	Discharge	3 months	7.792 *	1.461	0.000 *	4.900–10.684
		3 months	6 months	3.080 *	1.276	0.017 *	0.555–5.605
		6 months	12 months	1.896	1.080	0.082	−0.241–4.033
		12 months	18 months	2.176 *	1.006	0.032 *	0.185–4.167
		18 months	24 months	−2.600 *	0.900	0.005 *	−4.381–−0.819

* *p* < 0.05. MD—mean difference, SE—standard error, CI—confidence interval, MBI—modified Barthel Index, FIM—Functional Independence Measure.

## Data Availability

The datasets analyzed during the current study are available from the corresponding author upon reasonable request.

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
