# Peer review of "Changes in Long-Term Functional Independence in Patients with Moderate and Severe Ischemic Stroke: Comparison of the Responsiveness of the Modified Barthel Index and the Functional Independence Measure"

_ijerph, 2022, doi:10.3390/ijerph19159612_

Round 1

Reviewer 1 Report

An excellent study in which long-term functional changes were examined in patient with moderate and severe ischemic stroke.The moderate and severe groups were classified according to the NIHSS scores obtained at admission, and changes in functional independence were evaluated from discharge to 24 months after onset. 

The design of the study is very well founded, with a significant database of different stroke centers.

Metods clearly described.

The study results provide useful data for healthcare specialists who evaluate the functional status of stroke patients.

In conclusion, an excellent study. I don't find any issues not to be published in this form.

Author Response

In the introduction, review, and reference, the parts that need to be corrected were additionally supplemented.

thank you

Reviewer 2 Report

Dear Authors,

Thank you very much for sending the article titled: Changes in long-term functional independence in patients with moderate and severe ischemic stroke: comparison of the responsiveness of the Modified Barthel Index and the Functional Independence Measure. Generally, the paper is quite interesting to my mind, however, the authors should refer to the following statements:

- line 49 NIHSS - Please do not use abbreviations until the full name is given. It is described in 80 line. Additionally, I suggest not use the abbreviations in Abstract

- very poor state of the art. It would be great if authors write a few sentences about rehabilitation after stroke. For example in article titled: SPIDER as A Rehabilitation Tool for Patients with Neurological Disabilities: The Preliminary Research, JPM 2020, authors proposed a computer model for the evaluation patient’s condition and the rehabilitation progress. To improve manuscript quality I suggest cite this article and write a few sentences about methods used in patients rehabilitation after stroke. For example in line 273 authors write: ...our study suggests a tool that allows for a more sensitive assessment of the functional status of patients with severe stroke... There is place to describe methods which should be used for assessment of functional status.

- Please include in the article a number of Ethics Committee approval.

- line 131 (a flowchart should be inserted in one page)

- line 173, (Table 1) please include p-values between groups. Additionally, for example in Sex(%) Severe group we have 62(49.6), which means that coefficient of variation is 80%. This is evidence of a high volatility. As a rule, the reason is the large diversity of the research group. Could the authors refer to this value?

- line 177 (Table 2). What is the reason that the changes in the mean are statistically significant for the moderate group (except 18 month), while for the severe group the p-value is above 0.05?

- line 197 (Table 3). Please include p-value between groups. 

- line 200 (Figure 2). Please improve quality of this figure. It will be better if you insert a vector figure. You don't lose a quality

- line 221 (Table 4). Please change position of this table. It should be placed in Results section

- Please include before References the Abbreviation list

- References - Some names in literature are written in capital letters. Please correct it. 

Author Response

I have revised your request and presented your opinion.

thank you

Reviewer 3 Report

Thank you for submitting this paper. Unfortunately the evidence that you have presented does not offer anything new and I believe that you could improve the quality of your paper by including a more up to date discussion on neuroplasticity. You are in a unique position to ask the patient cohort how much rehabilitation they received post discharge, what they did to actually facilitate their recovery and how much help they were given. You obviously have access to an incredible data-base which would allow you to focus on 'best-intervention' times for different patient groups. If you could expand your discussion to include this then I believe you have an extremely valuable paper.

I have added my detailed comments on the attached pdf.

Author Response

(The authors gave the same response as above.)

Round 2

Reviewer 2 Report

Dear Authors, thank you for resending corrected article. 

Author Response

thank you for your sincere review.

Additional modifications have been made.

thank you

Reviewer 3 Report

Thank you for responding to my comments and for addressing them well. It is my opinion that the paper now reads better and that with some additional minor changes I have added to the pdf it will offer some additional useful information to the stroke literature.

Author Response

First of all, thank you for your sincere review.

I modified it as requested

thank you
